# A Web-Based Platform for the Automatic Stratification of ARDS Severity

**DOI:** 10.3390/diagnostics13050933

**Published:** 2023-03-01

**Authors:** Mohammad Yahyatabar, Philippe Jouvet, Donatien Fily, Jérome Rambaud, Michaël Levy, Robinder G. Khemani, Farida Cheriet

**Affiliations:** 1Department of Computer and Software Engineering, Polytechnique Montréal, Montréal, QC H3T 1J4, Canada; 2Department of Pediatrics, Faculty of Medicine, University of Montréal, Montréal, QC H3C 3J7, Canada; 3Department of Anesthesiology and Critical Care Medicine, Children’s Hospital of Los Angeles, Los Angeles, CA 90027, USA

**Keywords:** chest X-ray, machine learning, acute respiratory distress syndrome, pediatric acute respiratory distress syndrome, web-based platform

## Abstract

Acute respiratory distress syndrome (ARDS), including severe pulmonary COVID infection, is associated with a high mortality rate. It is crucial to detect ARDS early, as a late diagnosis may lead to serious complications in treatment. One of the challenges in ARDS diagnosis is chest X-ray (CXR) interpretation. ARDS causes diffuse infiltrates through the lungs that must be identified using chest radiography. In this paper, we present a web-based platform leveraging artificial intelligence (AI) to automatically assess pediatric ARDS (PARDS) using CXR images. Our system computes a severity score to identify and grade ARDS in CXR images. Moreover, the platform provides an image highlighting the lung fields, which can be utilized for prospective AI-based systems. A deep learning (DL) approach is employed to analyze the input data. A novel DL model, named Dense-Ynet, is trained using a CXR dataset in which clinical specialists previously labelled the two halves (upper and lower) of each lung. The assessment results show that our platform achieves a recall rate of 95.25% and a precision of 88.02%. The web platform, named PARDS-CxR, assigns severity scores to input CXR images that are compatible with current definitions of ARDS and PARDS. Once it has undergone external validation, PARDS-CxR will serve as an essential component in a clinical AI framework for diagnosing ARDS.

## 1. Introduction

Acute respiratory distress syndrome (ARDS) is a severe, even life-threatening condition, associated with respiratory failure, i.e., the inability of the lungs to fulfill their basic function of exchanging gases in the body. ARDS occurs in children and adults; its main causes include respiratory infection, aspiration, or trauma. The first description of ARDS as a separate disease was provided in 1967. Variability in the ability to identify ARDS causes difficulty in clinical trials. The Berlin definition introduced diagnostic criteria, such as acute onset, severe hypoxemia (lack of oxygen in the blood), bilateral diffuse infiltrates visible in chest radiography, and absence of any evidence of cardiac failure or fluid overload [1]. Despite intensive studies investigating ARDS (60,000+ articles found in PubMed), its mortality rate is still as high as 43% [2]. Among the survivors of ARDS, a significant portion experienced lasting damage to the lungs, especially in older patients. The Berlin definition grades the severity of ARDS as being mild, moderate, or severe. Table 1 illustrates the oxygenation criteria and mortality rates associated with these severity levels.

As seen in Table 1, considering the high mortality rate of ARDS and its rapid progression, early diagnosis of ARDS is vital. Furthermore, the mortality rate is directly associated with the severity of the syndrome. The risk benefit profile of therapies depends on ARDS severity, making early stratification of ARDS severity crucial for management. The Pediatric Acute Lung Injury Consensus Conferences (PALICC) [3,4,5] were organized to address pediatric ARDS (PARDS) specifications and give treatment and diagnosis recommendations. According to the most recent definition of PARDS, PALICC-2 [5], the criteria allow for new infiltrates in chest radiography, even if only a region within a single lung is affected. One of the main reasons for this change in diagnostic criteria was the lack of agreement in the interpretation of chest images between radiologists or between radiologists and intensive care practitioners on the presence of bilateral infiltrates, which are required in the Berlin standard. López-Fernández et al. showed that interobserver agreement for bilateral infiltrates and quadrants of consolidation in PARDS was “slight” (kappa 0.31 and 0.33) [6]. Sjoding et al. reported similar results, with interobserver reliability of ARDS diagnosis being “moderate” (kappa = 0.50; 95CI, 0.40–0.59). Hence, there is an urgent need to improve the reliability of Chest X-ray (CXR) intepretation in ARDS and PARDS to allow earlier diagnosis of the syndrome [7].

Several studies have applied machine learning (ML) and artificial intelligence (AI) approaches to analyze CXR images. One of the most common tasks reported in the literature is diagnosing pulmonary pathologies using chest radiography. Thanks to massive publicly available datasets, deep learning (DL) approaches have been broadly applied in medical pathology detection. However, there is as yet no dataset annotated with ARDS labels. Thus, few studies are found in the literature addressing the diagnosis of the syndrome.

To our knowledge, two papers present ML-based systems to identify ARDS in CXR images. The first one [8] proposed a method for detecting ARDS using a traditional ML approach based on hand-crafted features. The texture of intercostal image regions is considered as a discriminative feature for classifying samples. To highlight intercostal areas, a semi-automatic approach proposed by Plourde is utilized [9]. They succeed in reducing the inter-observer variability between clinicians in diagnosing PARDS. However, their approach is not automatic, and the rib segmentation step requires operator intervention. In the second work, an automatic ARDS detection and grading approach was proposed using a state-of-the-art DL model (Densenet) [10]. The authors first pretrained the model on public datasets (not containing ARDS samples) and then refined the model with a custom dataset consisting of ARDS-labeled images. Their approach performs well in diagnosing ARDS, but the model provides no evidence for the support system’s decisions. Thus, although it works well in analyzing ARDS cases, the model lacks interpretability, which is essential for a ML system to be used in clinical settings.

Recently, due to the COVID-19 outbreak, the research community has gotten more involved in computer-based analysis of chest X-ray images as one of the easiest and fastest ways to check for signs of the disease. Mobile Chest X-ray Analysis [11] and Chester [12] are prototype systems for CXR assessment developed using the aforementioned Densenet model, trained on the public Chest-Xray14 dataset [13]. Both systems provide evidence for the detected pathologies by means of saliency maps obtained using GradCam [14]. However, this can reveal areas that are irrelevant to the pathology being detected [15,16]. Thus, although these systems provide activation maps pointing out the references for the decisions, they are not sufficiently reliable to be used in clinics.

The main contributions of this paper are to create a tool for stratifying the severity of ARDS in CXR images and to build a web-based platform for external validation. The platform uses local information to classify X-rays based on the distribution of infiltrates in the different lung quadrants, and it provides a global severity score for the image that is applicable in both children and adults. The web-based platform, PARDS-CxR, can be used as a standalone tool, or it can be integrated with other ARDS analysis tools to offer a comprehensive approach for clinical use.

The following section first explains the details of the data collection used to train our DL model. Then, we describe the proposed DL model and its evaluation process, and we present the development of the Web platform. Section 3 presents the results of testing the ARDS assessment tool, and in Section 4, the strengths and drawbacks of our platform are discussed. Section 5 provides concluding remarks for this paper.

## 2. Materials and Methods

### 2.1. Methodology

This study contains four main phases, as illustrated in Figure 1. The end product is PARDS-CxR, the web-based application to detect ARDS. First, a substantial set of data is required to train the model. Existing public datasets do not include ARDS-labeled CXR images, so we created a new one. This data collection process is summarized in Section 2.2. Then, the proposed model must be trained on the CXR images. The model has two outputs associated with lung segmentation and ARDS classification, as explained in Section 2.3. The trained model is then tested on unseen data to be evaluated. Section 2.4, Section 2.5 and Section 2.6 detail the validation process. Finally, the model is uploaded to a server, and an interface is designed so the user can easily access it. The web application is addressed in Section 2.7.

### 2.2. ARDS Dataset

Collectively, the main publicly available CXR datasets provide around a million images with pathology labels [17]. This data motivated many researchers to employ AI techniques in this domain. However, no such datasets assign ARDS-specific labels to images. As our first step, we collected and annotated a dataset at Sainte-Justine Hospital, Montreal, Canada (CHUSJ), to address the lack of appropriate data. Our dataset comprises three data sources containing 373 CXR images. Ninety images and their corresponding labels came from a previous study by our team [8]. A further 100 images were taken from the Chest X-ray14 dataset [13] and relabeled by clinical experts (JR, ML) in the hospital. Another 183 images were provided by the PARDIE study, a multi-national study that prospectively gathered chest X-ray images of children with ARDS [6]. For each image, labels were associated with the four lung quadrants obtained by splitting each lung into upper and lower portions. We refer to each quadrant by its position: left upper (LR), left lower (LL), right upper (RU), and right lower (RL). According to the Berlin definition [1], visible bilateral infiltrates are a mandatory criterion for a case to be categorized as ARDS. Two intensivists from CHUSJ assessed the presence of infiltrates in each quadrant. A sample was included in the dataset only if the clinical observers reached a consensus on the labels. In addition, 138 CXR images were taken from the Montgomery dataset to represent the normal class [18]. These samples were labeled as non-ARDS if agreed by the clinical experts. By dropping images with disagreements in their labeling, our final ARDS dataset consisted of 356 images, of which 134 meet the bilateral infiltrates criteria in the Berlin definition [1].

### 2.3. Joint Segmentation and Classification Model

In computer-based diagnosis approaches, it is common to use segmentation ahead of classification to determine the region of interest. Lung segmentation separates the lung areas from the thoracic tissues surrounding them and is the primary image analysis step in many clinical decision support systems. Generalization to new datasets is a difficult challenge in the analysis of chest radiography. In that respect, segmentation is considered a strategy to limit the impact of specific imaging devices and settings, since it restricts the feature extraction to the lung fields and removes the effect of the image background [19,20]. However, serial usage of segmentation and classification propagates the segmentation error into the classification network. Dense-Ynet is a convolutional network that takes advantage of Densenet, Y-net, and U-net models to do both tasks simultaneously in a joint segmentation–classification model. The backbone of the network used in this study is our previously developed Dense-Unet [21]. Dense-Unet is a segmentation model in which dense connections between the feature maps in various layers facilitate the information flow throughout the model, letting designers choose a configuration with a small number of training parameters. Our proposed Dense-Ynet takes advantage of automatic feature extraction from both the original and segmented images (Figure 2). The model has two outputs and is trained using two loss functions: the lung segmentation loss and quadrant classification loss. The model works based on the convolution operation. A convolution is a mathematical operation that filters the information of its input and creates feature maps. An inevitable effect of the convolution operation is to change the dimensions of the feature maps. To tackle this issue, upsampling and strided convolution operations are used to ensure that feature maps coming from different layers can be concatenated. Squeeze and excitation (SE) blocks [22] are also used after each convolution layer to improve the representational power of the blocks by recalibrating the features. The key strengths of Dense-Ynet are use of lung segmentation in its architecture, specialized connectivity, which enable better generalization, and prediction of local labels for each image.

To reach the final decision based on the Berlin definition, we must test for existing bilateral infiltrates. To that end, a simple logical operation in Equation (Equation 1) combines the predictions of each quadrant to check this condition:(1)$$P_{ARDS}=\left(P_{RL}\vee P_{RU}\right)\wedge \left(P_{LL}\vee P_{LU}\right)$$

$P_{RL}$, $P_{RU}$, $P_{LL}$, and $P_{LU}$ are the prediction labels for the right lower, right upper, left lower, and left upper quadrants, respectively. $P_{ARDS}$ is the inferred ARDS label, and ∨ and ∧ are logical *or* and *and* operations. The equation states that, if at least one quadrant is involved on each side, the case is recognized as (P)ARDS.

### 2.4. Experimental Design

In this work, 267 images of the ARDS dataset are used to train the Dense-Ynet model. In addition, 35 images are used to validate the training process. For the testing stage, 54 images previously unseen by the network are used. The algorithm is evaluated with the five-fold cross-validation strategy. Cross-validation is a method that tries various training and testing data combinations to confirm the reported results’ reliability. Data augmentation is a technique to enrich the training data by generating new images from the current training set. For this purpose, we use basic image processing techniques, such as random rotation, cropping, shifting, horizontal flipping, and intensity changing. The rectified linear unit (ReLu) activation function introduces non-linearity to network blocks. The Sigmoid function provides valid labels between zero and one in both the segmentation and classification output layers. Adam is the optimizer used for updating the model weights during training. To reach the optimal configuration, a set of hyperparameters must be explored to find the best model structure and training policy. The Web platform (see Section 2.7) employs six Dense-Ynet instances, corresponding to the best hyper-parameters sets. Using an ensemble approach, the final result presented to the user combines the values received from the individual models.

The PARDS-CxR application detects lung quadrants consolidation, and the final ARDS label is derived from the quadrant predictions using Equation (Equation 1).

### 2.5. Scoring Scheme

To analyze the severity of ARDS in CXR images, a scoring scheme is proposed based on the number and the position of affected lung quadrants (see Table 2). The scheme is compatible with the Berlin definition, in which existing bilateral infiltrates are an essential criterion for ARDS diagnosis in chest radiography.

Giving scores is important from two points of view. First, the score represents the severity of the diffused infiltrates throughout the lungs. Second, reporting disease severity helps clinicians follow appropriate treatment protocols or triaging. This type of system has been proposed for the Murray Lung Injury Score, as well as as part of the recently proposed RALE score in adult patients with ARDS.

### 2.6. Evaluation Metrics

Evaluation metrics are measured from the algorithm’s performance on unseen test data to assess the approach. There is no metric representing the total capacity of the PARDS-CxR platform. However, we use a set of performance metrics to provide a complete overview of the model’s operation. A confusion matrix quantifies the ability of the classifier to detect each class separately. It gives detailed measures comparing the actual and predicted labels, as shown in Figure 3.

The elements of the confusion matrix, namely, the true positive (*TP*), true negative (*TN*), false positive (*FP*), and false negative (*FN*) values, serve to calculate several assessment metrics as follows:(2)$$Accuracy=\frac{TP+TN}{FP+FN+TP+TN}$$
(3)$$Precision=\frac{TP}{FP+TP}$$
(4)$$Recall=\frac{TP}{FN+TP}$$
(5)$$F1=\frac{2\times Precision\times Recall}{Precision+Recall}.$$

The Accuracy metric represents the overall correctness of a classification algorithm. It cannot fully express the model performance, however, especially in the case of unbalanced testing data. Precision and Recall reveal the model’s performance in discriminating between the different classes. Precision represents how precise the model is in identifying the target (positive) class. Specifically, it points out what portion of cases predicted as positive are really ARDS cases. On the other hand, the Recall value shows what proportion of predicted ARDS cases are actually labeled as ARDS. These two metrics have a complementary role in describing the model’s behavior. The F1 score, derived from Precision and Recall values, is a single metric to quantify the algorithm’s performance.

The receiver operating characteristic (ROC) curve illustrates the diagnostic capacity of a system by comparing true positive and false positive rates as the discrimination threshold (applied at the network’s output layer to decide between the two classes) varies. The area under the ROC curve (AUROC) represents the discriminatory power of the classifier.

### 2.7. Web-Based Platform

We designed a web-based platform to facilitate the diagnosis of ARDS in CXR images by medical professionals. The platform is intended as a tool to provide a second opinion to clinicians, but no direct medical use is recommended until medical professionals validate the tool using external data. The PARDS-CxR platform takes advantage of six Dense-Ynet instances to provide scores for each input image. The scores are given based on the number and combination of affected lung quadrants as explained in Section 2.5. A global score is assigned by combining the outputs from the model instances. In addition, the application provides accurate lung segmentation maps, which are helpful in AI-based analysis of CXR images.

The web application utilizes the *React* library to create a user-friendly and interactive user interface (UI) for delivering the specified services. The library enables efficient code writing and makes it easier to manage, refine, and integrate the application with other tools. The platform supports both English and French languages and has two main modes for ARDS definitions for adults (Berlin) and children (PALICC-2). The difference between the modes is that, when using PALICC-2 mode, the platform requires two input images. The application response includes segmentation maps, severity scores (local and global), and an interpretation based on the definition.

Although the deep models are trained using graphical processing units (GPUs), the evaluation model does not require a GPU and can process the results in 2–3 s. Thus, the running bottleneck could be the network connection speed. The application is capable of storing data and providing log files, but this feature is currently disabled and will be activated when the validation protocol is approved. The PARDS-CxR platform is detailed further in Section 3.3.

## 3. Results

### 3.1. Quadrant-Based Classification

The PARDS-CxR web-based platform uses Dense-Ynet as the joint segmentation-classification model. In classification, the model predicts four labels associated with lung quadrants, as explained in Section 2.3. The platform uses an ensemble of six Dense-Ynet model instances with different training and model structure configurations. Regarding model structures, we experimented with different channel depths in convolution blocks, loss functions, weights for merging loss functions, activation functions, and initial network weights. For the training configurations, we varied several hyperparameters, namely, the learning rate, training batch size, augmentation probability, and stopping criterion.

Figure 4 shows the confusion matrix of the ensemble of models. To merge the results from the model instances, a hard voting strategy is employed based on the labels predicted independently by the models. To be precise, each model is trained separately with its specific configuration. The testing is also done independently, and if at least three models decide that an image is an ARDS case, the combined result is positive. By combining models with various configurations, the intrinsic biases of each one to accept or reject an image as ARDS are balanced in the ensemble output. Thus, the final performance improves compared to any individual model.

Table 3 compares the classification performances of the Dense-Ynet instances in terms of the four assessment metrics seen previously. Some of the listed models achieve higher precision, while others reach better recall values. By combining the predicted labels provided by these models, the ensemble algorithm achieves the highest F1 score, representing the best compromise between precision and recall. Indeed, ensembling the models does not outperform every one in terms of Precision and Recall, but the final F1 and accuracy values improve.

In this paper, the problem of ARDS diagnosis is based on the classification of lung quadrants. Thus, the task can also be considered as a multi-label classification problem. Figure 5 shows the ROC curves of all quadrants’ predictions for the Dense-Ynet instances, i.e., the ROC curves associated with the binary classification of the lung quadrants, regardless of their positions. The AUROC metric is not directly related to the system’s performance in ARDS diagnosis, but the misclassification of one lung quadrant may cause an error in classifying the image as a whole.

### 3.2. ARDS Severity Prediction

As seen in Table 2, the application determines the severity of ARDS in CXR images based on the number and combination of affected lung quadrants. The platform provides a global score for each input image by taking the average of the scores from each model. CXR images are then categorized into one of three severity grades based on the predicted scores: non-ARDS, mild ARDS, and severe ARDS. The platform’s effectiveness in determining ARDS severity is illustrated in Figure 6. The three-class confusion matrix shows that the approach can detect ARDS and discriminate between mild and severe states of the syndrome.

### 3.3. PARDS-CxR, the Web-Based Platform

Our web application is currently loaded on a web server at CHUSJ and is accessible at the address (https://chestxray-reader.chusj-sip-ia.ca, accessed on 15 January 2023). The process of training and testing the deep model was programmed in Python using the PyTorch library [23]. The training process and hyperparameter search were executed on GPU, as they required intensive parallel computing. The trained model was then transferred to CPU to evaluate new images; thus, no graphical processor is necessary on the server to run the application. The graphical user interface was written in JavaScript and is compatible with various internet browsers on the client side. No data are kept on the server side, and the application output image is available to store in the user’s local storage. The user interface works in English and French, and CXR images can be uploaded using the menu option or drag-and-drop (see Figure 7).

The application bases itself on the most accepted definitions for ARDS and PARDS. Based on the Berlin definition, the presence of bilateral infiltrates in chest radiography is a criterion manifesting the existence of ARDS [1]. The platform processes the image and displays its decision by providing a percentage associated with the level of infiltration in each quadrant (Figure 7). A global percentage is also given based on infiltrate levels of infiltrates in quadrants and their combination as in Table 2. This value represents the severity of ARDS in the input image. An image with a global percentage above 60% is interpreted as an ARDS case, since, based on the proposed severity scoring system, infiltrates should be diffused through both lungs. Reporting each quadrant’s involvement is necessary, since it gives the rationale behind the global severity measure. As seen in Figure 7, a segmentation map highlighting the lung segments is also provided.

Identifying progression of ARDS is also possible, as two images taken at different times can be compared by the system. Additionally, an example of CXR image comparison is displayed in Figure 8.

## 4. Discussion

The proposed DenseY-net is a joint segmentation–classification model that diagnoses (P)ARDS based on lung quadrant-level classification. The results show that the model can accurately classify quadrants and, consequently, the entire input image. This labeling strategy offers a reasoning framework for decision-making and incorporates an interpretability feature into the platform. Ensemble modeling is used to combine the outcomes from six model instances. PARDS-CxR can also do lung field segmentation, which is a necessary element in many decision support systems. Our approach performs well in detecting the severity of ARDS by giving a score to each input determined by the number and position of affected lung quadrants. This makes the model compatible with both ARDS and PARDS definitions.

A few large chest radiography datasets are publicly available for the research community [13,24,25]. A key benefit of deep learning is its capacity to analyze and learn features from a substantial amount of data. Therefore, it is unsurprising that several ML researchers have investigated CXR image analysis in various contexts. However, important limitations of these datasets make them unsuitable for developing dependable systems for the hospital setting. Indeed, most of the data are annotated using clinicians’ notes processed by natural language processing (NLP) techniques [26]. This leads to erroneous labeling of a portion of the images. For example, a 10% error is reported for Chest X-ray 14 [13], even though it is one of the most frequently used CXR datasets. The clinical review in [27] reveals an even higher rate of data labeling errors in that dataset.

Although adding some level of noise to the training inputs can improve a deep model’s performance, biases and extensive labeling errors will decrease the model’s accuracy. This could be a reason for the relatively poor generalization ability of deep models when confronting new samples from other data sources. Furthermore, available samples are annotated for a limited number of pathologies. Public CXR datasets cover between 14 and 18 chest pathologies, but these do not include ARDS or PARDS. To address this constraint, we collected our own CXR dataset from three different sources and annotated it for PARDS at CHUSJ. This dataset was labeled at the lung quadrant level, and the lung fields were manually identified in each image to establish a segmentation ground truth. The resulting dataset contains 356 CXR images, including 134 that meet the bilateral criteria for ARDS. Annotating data is costly in the clinical field, even more so considering that the DenseY-net model needs lung maps and quadrant-level ground-truth labels. Consequently, our ARDS dataset is relatively small. Nonetheless, our model is designed in such a way as to train adequately on small datasets. The specialized connectivity within the model allows for the creation of a lighter model with shallower intermediate feature maps, resulting in a smaller number of training parameters. A model with fewer parameters is more appropriate for training with small datasets. The algorithm was assessed on our own dataset, as explained in Section 2.4. A bigger dataset could increase the generalization capacity of the model ensemble. Moreover, external validation of the platform using data from various health centers will make it more reliable as a tool for prospective clinical research. Thus, as next steps in the web application’s development, external validation and improving interpretability are two major points, since both are necessary to turn the platform into a practical tool in clinics.

Moreover, according to the (P)ARDS definition, co-occurrence of detectable infiltrates in CXR and hypoxemia is necessary when no evidence of cardiogenic pulmonary edema is observed. Thus, although the presence of infiltrates in chest radiography is known as the most limiting factor for diagnosing ARDS, meeting other criteria is a requisite. The Clinical Decision Support System (CDSS) lab at CHUSJ has the capacity to investigate other ARDS diagnosis criteria, including cardiac failure and hypoxemia. Le et al. have employed NLP techniques and ML algorithms to detect cardiac failure in children [28]. Sauthier et al. have developed a method to accurately estimate Pao2 levels using noninvasive data [29]. Integrating the tool proposed in this study with other works will lead to a system giving comprehensive ARDS diagnoses. Sufficient electronic medical infrastructure is available in the PICU of CHUSJ to facilitate the flow of data from various sources [30]. By accessing data from clinical narrative analysis, measuring oxygenation indices, and detecting infiltrates in CXR images, it will be possible to make clinical decisions in real time. Therefore, an important objective for our team will be to implement an ARDS diagnosis package at CHUSJ, integrating all these criteria and data sources.

The strength of this study lies in the development of an algorithm that, in comparison to existing approaches, is more interpretable and automated and is compatible with existing ARDS definitions. Unlike an earlier ARDS diagnosis method proposed by our research team [8], the DL-based approach used in this application does not need any interaction from clinicians or operators to guide the algorithm. The novel model provides an end-to-end process that is simple for the user and provides the diagnotic outputs instantaneously. Recently, Sjoding et al. [10] proposed annother automatic algorithm for detecting ARDS in CXR images. However, their approach lacks explainability, i.e., the system’s decisions are not supported by further information. By contract, since PARDS-CxR detects infiltrates in each lung quadrant, the basis for the decision is integral to our method. This strengthens the platform’s reliability, since the user can reject or accept the decision by observing the delivered explanation. In addition, the proposed approach is compatible with both PARDS and ARDS definitions [1,3], as the scoring scheme used translates to a disease severity level. At present, the main limitation of our algorithm is its lack of external validation. Indeed, its development relied on a limited number of CXR images with a single team annotating them. For this reason, we have implemented the algorithm on a web platform to allow researchers to conduct validation studies.

## 5. Conclusions

This work has described a deep learning method and web-based platform for diagnosing acute respiratory distress syndrome (ARDS) from chest X-ray (CXR) images. The platform uses an ensemble of novel Dense-Ynet networks that can accurately detect lung infiltrates in different quadrants and combine this information to detect ARDS and grade its severity. This approach ensures that our tool is compatible with various ARDS definitions in both adults and children. Following feedback from clinical researchers during a validation phase, the platform will be integrated into a complete clinical decision system for ARDS. The tool presented here will serve as the CXR analysis component within an AI-based framework that will monitor other factors, such as hypoxemia and occurence of cardiac arrest.

## Figures and Tables

**Figure 1 diagnostics-13-00933-f001:**
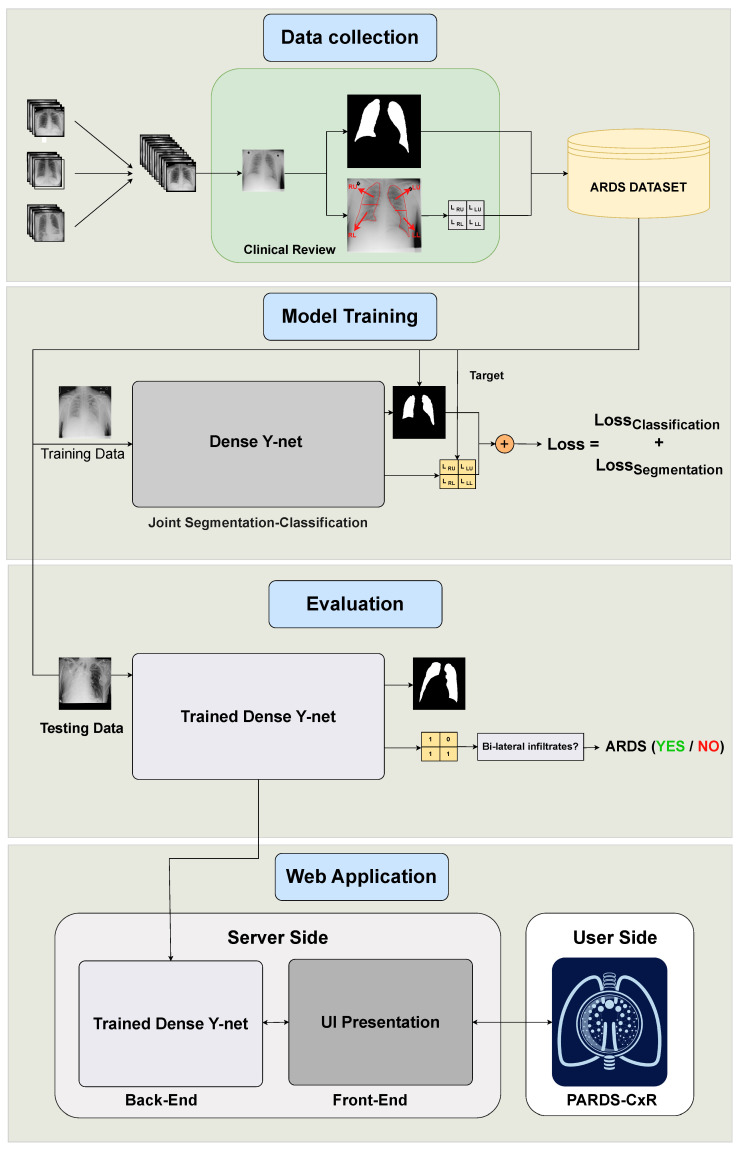
Organization of our study into four main phases. The data are collected from several data sources and annotated at Saint-Justine Hospital, Montreal, Canada. The DL model is trained using quadrant-level labels and lung segmentation maps. It is then evaluated on a set of previously unseen images; both the classification and segmentation performances are assessed. Finally, a web-based platform is designed and made available through the internet.

**Figure 2 diagnostics-13-00933-f002:**
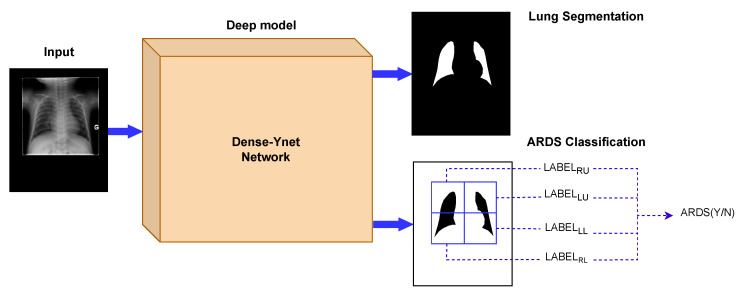
The Dense-Ynet model takes advantage of the interaction between the segmentation and classification tasks by performing them simultaneously. The features from the original and lung-segmented images are concatenated and utilized to classify ARDS cases.

**Figure 3 diagnostics-13-00933-f003:**
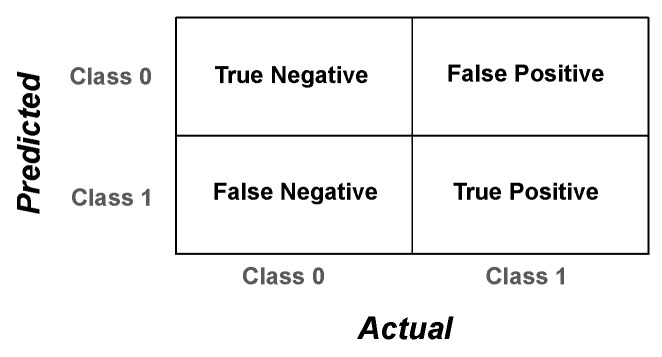
Confusion matrix for a binary classification problem. The matrix contains four elements that, together, evaluate the system’s predictions versus the real labels.

**Figure 4 diagnostics-13-00933-f004:**
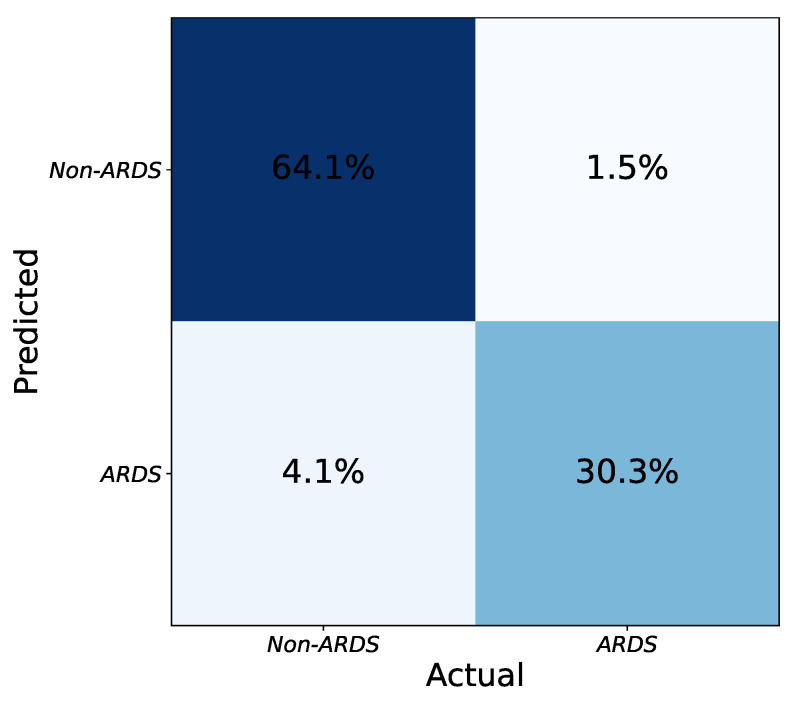
Final confusion matrix obtained from the combination of network instances using hard voting. The numbers (percentages) are obtained by taking the average of several tests (five-fold cross validation).

**Figure 5 diagnostics-13-00933-f005:**
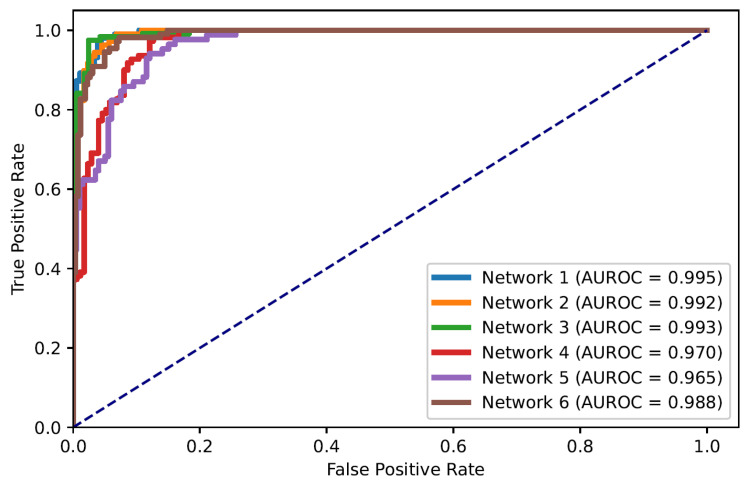
ROC curves for classification of lung quadrants regardless of their position in the lungs.

**Figure 6 diagnostics-13-00933-f006:**
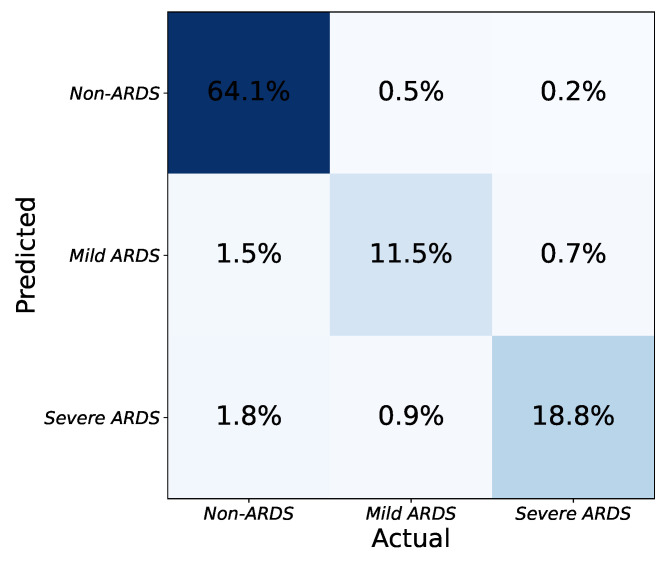
Confusion matrix for classification of ARDS severity with three levels (none, mild, severe).

**Figure 7 diagnostics-13-00933-f007:**
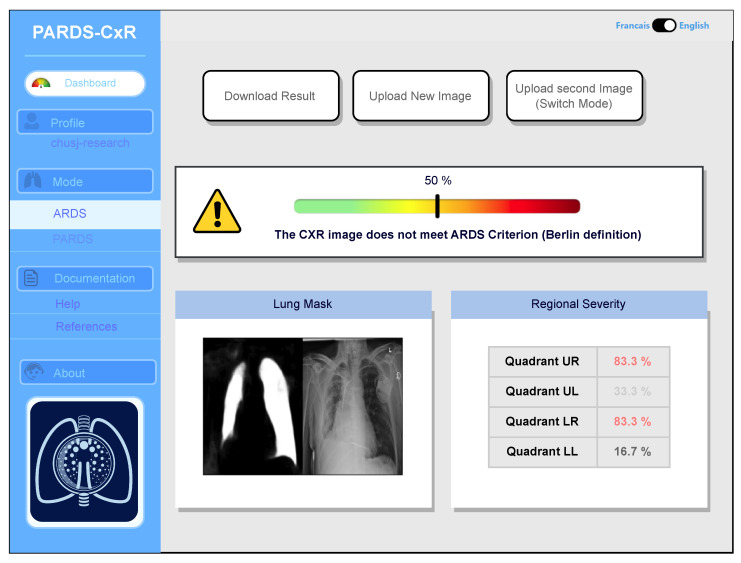
Main interface of the PARDS-CxR web application. In the standard mode, a single CXR image is analyzed according to the Berlin definition.

**Figure 8 diagnostics-13-00933-f008:**
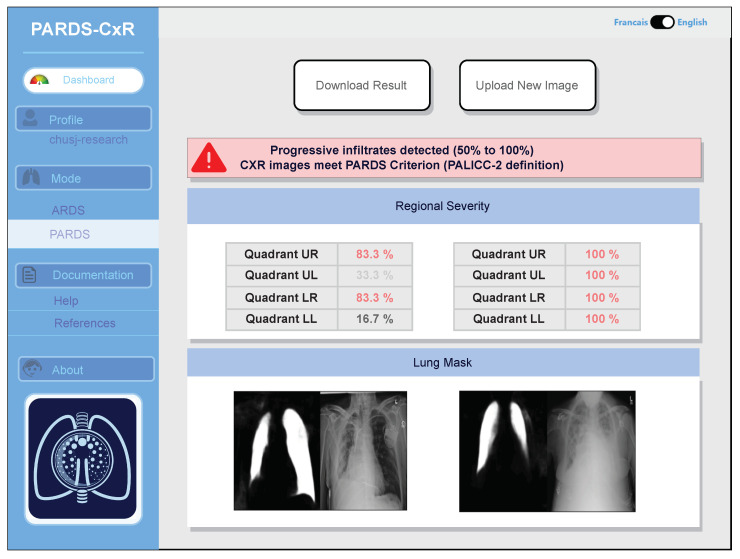
PARDS-CxR interface in image comparison mode. The platform can analyze two CXR images to detect ARDS progression based on the PALICC-2 definition.

**Table 1 diagnostics-13-00933-t001:** ARDS severities in the Berlin definition and associated oxygenation levels and mortality rates [1].

Severity	PaO_2_/FiO_2_	Mortality
Mild	200–300	27%
Moderate	100–200	32%
Severe	≤100	45%

**Table 2 diagnostics-13-00933-t002:** Severity scoring scheme based on affected lung quadrants.

Affected Quadrants	Score	Severity
4 quadrants	5	Severe
3 quadrants	4	
2 quadrants (Different sides)	3	Mild
2 quadrants (Same side)	2	Non-ARDS
1 quadrant	1	
No affected quadrant	0	

**Table 3 diagnostics-13-00933-t003:** Evaluation of the six models and the result of their combination (ensemble model) for classification.

Model	Accuracy	Recall	Precision	F1
Network 1	92.95%	88.45%	91.99%	90.19%
Network 2	93.54%	96.41%	84.37%	89.99%
Network 3	92.04%	94.42%	87.89%	91.03%
Network 4	92.96%	100.0%	83.33%	90.91%
Network 5	87.32%	100.0%	74.29%	85.25%
Network 6	88.74%	80.01%	80.01%	80.02%
Ensemble model	94.35%	95.25%	88.02%	91.49%

## Data Availability

Access to data can be requested from Philippe Jouvet. Specific institutional review board rules will apply.

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
