# Peer review of "A Web-Based Platform for the Automatic Stratification of ARDS Severity"

_diagnostics, 2023, doi:10.3390/diagnostics13050933_

Round 1
Reviewer 1 Report
This work is about a web-based platform designed in which analysis of CXR images is established to diagnose ARDS. Due to the outbreak of COVID-19, the diagnosis of pneumonia or ARDS by AI became an important issue. The strength of this study is the development of the algorithm, in comparison to existing approaches, that is interpretable, automatic, and flexible to ARDS definitions. The ROC curve showed good results in prediction. It is an interesting work and gives good teaching points about the machine learning model for readers.
Author Response
Dear reviewer,
The authors would like to thank the reviewer for the careful review of the manuscript and positive comments especially the following: “The strength of this study is the development of the algorithm, in comparison to existing approaches, that is interpretable, automatic, and flexible to ARDS definitions. The ROC curve showed good results in prediction. It is an interesting work and gives good teaching points about the machine learning model for readers.”
Best Regards,
Reviewer 2 Report
Overall paper is well-written and presented. But, still need to work on some reviews to publish the paper.
1. The contributions of the authors are confusing and unclear.
2. What's the difference between the proposed and existing models?
3. A lot of work has been done in similar technology and approaches, what's the perspective of this study and what are the advantages of your model?
4. Author mentioned that the ARDS dataset consists of 267 images that have been used to train the model Dense-Ynet. Is it possible to train properly a deep learning model where the Keras model is trained by millions of images? In that case, how do authors handle the overfitting issues?
5. Did the author conduct any statistical analysis to confirm the model's robustness? I strongly recommend this.
6. Figures 1-10 are hazy and unacceptable.
7. Rewrite the conclusion section, It not properly concluded the paper.
8. Try to use recent studies as citations (2015-2023).
9. A grammatical proofread is needed.
Round 2
Reviewer 2 Report
The authors have addressed all of the comments successfully. I have no further comments regarding this paper.
Author Response
Thank you again for your review suggestions